# Turbulent Length-scales in a Fast-flowing, Weakly Stratified, Strait: Cook Strait, New Zealand

Craig L. Stevens[1,2]

[1] National Institute of Water and Atmospheric Research, Greta Point, Wellington, 6021, New Zealand.
[2] Department of Physics, University of Auckland, New Zealand

*Correspondence to*: Craig Stevens (craig.stevens@niwa.co.nz)

**Abstract.** There remains much to be learned about the full range of turbulent motions in the ocean. Here we consider turbulence and overturn scales in the relatively shallow, weakly stratified, fast-flowing tidal flows of Cook Strait, New Zealand. With flow speeds reaching 3 m s$^{-1}$ in a water column of ~300 m depth the location is heuristically known to be highly turbulent. Dissipation rates of turbulent kinetic energy $\varepsilon$, along with the Thorpe scale, $L_T$, are described. Thorpe scales, often as much as one quarter of the water depth, are compared with dissipation rates and background flow speed. Turbulent energy dissipation rates $\varepsilon$ are modest but high for oceans, around $5\times10^{-5}$ W kg$^{-1}$. Comparison of the buoyancy-limit Ozmidov scale $L_{Oz}$ suggest the Cook Strait data lie for the majority of the time in the $L_{Oz}>L_T$ regime, but not universally. Also, comparison of direct and $L_T$ -based estimates of $\varepsilon$ exhibit reasonable similarity.

## 1    Introduction

It is well-established that turbulent mixing in the ocean is intermittent and patchy (see Waterhouse et al., 2014 for a synthesis). Thus, there is substantial benefit in seeking out extreme conditions to fully capture the global energy budget. Tidal motion, through one pathway or another, drives significant mixing in the ocean. While is it understood that this mixing can influence ecological functioning (e.g. Scott et al., 2010; Koch-Larrouy et al., 2015), knowledge gained in shallow coastal situations is also applicable in deeper ocean conditions (e.g. Dale and Inall, 2015). Here we consider these issues in the fast flows of a large tidally-driven passage, Cook Strait, a situation that couples a relatively large vertical extent with substantial inertial forcing.

In a 1999 paper reviewing the first shear probe measurements of oceanic turbulence Stewart and Grant (1999) described the flows in Seymour Narrows (Discovery Passage, Canada) as sustaining Reynolds Numbers (Re) amongst the "largest in the universe". True or not, it is a useful benchmark and discussion point. There is a tendency to ignore Reynolds number in geophysical flows as they are typically so very large, primarily because of the length-scales involved. Cook Strait has comparable flow speeds to Discovery Passage but is around four times the depth, and so suggests a larger bulk Re. From the diapycnal diffusion perspective, despite this highly turbulent large-scale flow, stratification clearly persists through the strait (Stevens, 2014).

Of practical concern here is the amount of kinetic energy lost from the system via dissipation (i.e. the rate of dissipation of turbulent kinetic energy, $\varepsilon$) as this plays an important role in adequately simulating ocean systems where there

is a high dynamic range of variability. This then informs quantification of turbulent diapycnal diffusion which is a balance of turbulent overturning against a stably-stratified background as characterised by the buoyancy frequency squared $N2=(g/\rho)(d\rho/dz)$. Wesson and Gregg (1994) set the scene for the research theme surveying turbulence quantities in the exchange-dominated Straits of Gibraltar where they were able to quantify key turbulence parameters as driven both by internal

shear and boundary mixing.

        Mater and Venayagamoorthy (2014) lay out a pictorial representation of length-scales in stratified turbulence. The observed Thorpe overturning scale, $L_T$, is a relatively measurable quantity associated with ocean structure and can be considered the energy baring scale. This is constrained by the Ozmidov scale ($L_{Oz}=[\epsilon/N^3]^{1/2}$) that identifies the limits to growth of eddies and also the Kolmogorov length-scale ($L_K=[\nu^3/\epsilon]^{1/4}$, $\nu$ is kinematic molecular viscosity) where turbulent fluctuations

are absorbed by viscous damping forces. It is common to seek to relate the observable $L_T$ to mechanistically relevant quantities like turbulent kinetic energy, and its dissipation rate $\epsilon$ (e.g. Dillon, 1982; Mater et al., 2013). This enables $\epsilon$ to be estimated from a combination of relatively achievable measurements (Mater et al., 2015; Scotti, 2015) explore the veracity of this long-used approach in a variety of conditions. Typically, this has been examined in the deep ocean and so biased away from the more energetic conditions.

The present paper uses microstructure and overturn data to report on the stratified boundary layer response and mixing in the unique situation of Cook Strait as an aid extending our knowledge around oceanic turbulence. It is instructive to compare Cook Strait with other straits of note (Table 1) as it is essentially oceanic, and so relatively weakly stratified, with the g′ in Table 1 being a maximum as observed through an annual cycle. The table includes representative estimates of the Reynolds number and a bulk Richardson number ($Ri= g′h/\Delta u^2$) where $\Delta u$ is the top-bottom velocity difference (Ri~<1 implies weak

stratification). The remarkable aspect for Cook Strait is its tidally induced currents and so provides a useful location (Figure 1) because of the very fast tidal flows in reasonably deep water. A number of questions arise: (i) Do we actually observe high dissipation rates? (ii) How does the Thorpe Scale compare with the Ozmidov Scale? (iii) Following from this, can a fixed ratio be assumed and so allow estimation of $\epsilon$? (iv) How does the turbulence compare with other straits?

**2      Location and Sampling**

        Cook Strait, the channel separating New Zealand's North and South Islands, connects the eastern Tasman Sea to the Western Pacific at 42 S (Figure 1). At its narrowest point it is 22 km across, with 210 and 350 m average and maximum depths, respectively. Its fast-flowing tidal currents have been the focus of a number of studies, including the notable observation that the semidiurnal tide is around 140 degrees out of phase when considering the opposite ends of the Strait

(Heath, 1978). This phase difference drives substantial flows, reaching as high as 3.4 m s$^{-1}$ during spring tides (Stevens et al. 2012).

Background velocity data come from two instrumented moorings deployed at the "third points" across the narrows (Figure 1) for a period spanning two years, in two deployments, starting in August 2010 and continuing through until September 2012. Each mooring contained an upwards-looking Teledyne-RDI 75 kHz ADCP mounted in a Flotation Technologies syntactic foam float and moored with 600 kg of iron and 10 m of chain. The ADCPs logged at 10 minute intervals, sampling into 8 m depth bins. Each float contained a Seabird microcat (SBE 39) conductivity/temperature/depth sensor placed beneath the ADCPs which sampled at 5 minute intervals. This enabled comparison with satellite-derived sea surface temperature for the centre of the strait. With such high tidal flow rates it is not possible to adequately moor instrumentation near the surface as the mooring is "knocked down" meaning that near-surface data are not observed during high flows.

Microstructure profiles were recorded with a VMP500 (Vertical Microstructure Profiler - Rockland Oceangraphic, Victoria Canada) instrument. This free-fall, loose-tether package supported two shear probes, two fast thermistors, accelerometers and a Seabird Electronics (SBE) conductivity and temperature sensor-pair. Thirty-four profiles were collected using the 14 m twin-hulled jet-boat Ikatere during a number of expeditions from 2010-2012 but the bulk come from a 12 day period in 2012. The timing of the profiles during the 2012 sampling is shown in Figure 2. It is difficult to capture extended periods of contiguous sampling because a vessel suitably manoeuvrable to conduct the experiments is prone to weather limitations. Sampling over three days in 2012 centered on periods spanning northward, turning and southward tidal flows (Figure 2).

The profiler captures temperature and conductivity data, however this sensor-pair is un-pumped (to reduce vibration contamination of the shear probe) and so has a slow actual conductivity response and is relatively sensitive to response-time mismatch induced spiking. A fast response conductivity sensor was included in the measurements which gets around the response issue but had its own idiosyncrasies due to fouling and will not be examined here. Correcting un-pumped salinity estimates is becoming more common with ocean glider applications (Timmermans and Winsor, 2013), however the present profiling application is a more rigorous challenge. Being a derivative quantity, $N^2$ emphasises any spikes or noise. The bulk temperate-salinity relationship for the region is relatively well-ordered and so this enables density for each profile to be calculated using the high-quality temperature and the bulk T-S relationship (for that profile). While this would not be particularly reliable for absolute density estimation, it is sufficient to generate an estimate of the buoyancy frequency squared $N^2$. The density profile contains fine-scale overturns and this also results in a challenge for $N^2$ estimation. Mater et al. (2015) review methods for calculating $N^2$, and here the patch-average $N^2$ is used based on a density-sorted profile. The removal of salinity spikes from the original profile data was found to have the greatest impact on the $N^2$ estimation.

The microstructure data were recorded using a pair of orthogonally mounted shear sensors. The shear data were recorded at 512 Hz and processed to resolve the dissipation rate $\varepsilon$ (Wolk et al., 2002; Macoun and Lueck, 2004). This involved first de-spiking to remove spurious transient records, most likely due to encountering biological organisms. The dissipation

can then be determined from the integration of shear. However, before this is calculated the useful limit of the data needs to be determined. Unlike many microstructure applications, there is a high signal to noise ratio. What challenges these data is profiler vibration (Wolk et al., 2002). The profiler also samples package motion using a triaxial accelerometer and this provides a cut-off point in the useful shear data, beyond which the spectrum is padded with the Nasmyth spectrum (Macoun and Lueck,

2004). The data were separated into dissipation rate estimates from each sensor using 5m depth bins. The requirement is that the profiler be passing through the water steadily over the period of any given bin. Vertical speed is resolved from the pressure sensor so that conversion to wavenumber requires reliable velocity estimation (Wolk et al., 2002). Figure 3 shows the profiler drop speed and its variability reflects the degree of vertical turbulent motion which reached as much as 0.1 m s$^{-1}$. The upper portion of the water column includes an acceleration period and sometimes wave effects are apparent. Deeper down it is clear

that there is variability in the character of the drop speed variations, although over periods longer than that required for the 5 m vertical bins. The shear spectrum was generated for each depth bin and then compared with a pseudo shear spectrum generated from the accelerometer data. The cross-over point allowed identification of the noise limit in the shear spectrum above which the signal was replaced this with the Nasmyth model spectrum. With the generally high dissipation rates this was not a particularly significant correction.

15        Having resolved $\varepsilon$ and $N^2$ this then enables a number of derived quantities to be calculated. The Ozmidov scale $L_{Oz}=(\varepsilon/N^3)^{1/2}$ identifies the upper bound at which eddies should "feel" the stratification. One might expect overturns, as identified using the $L_T$, to be equal to, or smaller than $L_{Oz}$. Dillon (1982) observed the ratio to be $L_T/L_{Oz}$ =0.8. This calculation struggles in regions of weak stratification where locally-small $N^2$ results in a very large scale. This makes sense as weak stratification fails to retard turbulence. However, it can also lead to non-physical outcomes, as the scale will eventually

exceed water depth. The vertical (~diapycnal) diffusivity $K_z$ is commonly calculated as $K_z=\Gamma\varepsilon/N^2$ with $\Gamma$=0.2 an assumed constant. While convenient, there is a good deal of evidence to suggest that $\Gamma$ is not constant – for example Bluteau et al. (2013) suggested resulted in an order of magnitude over-estimation of mixing rate. This will be returned to in the Discussion.

        Given the nature of the salinity structure, as with Wesson and Gregg (1994) and others, we use the more precisely-known temperature to define overturns. The Thorpe scale $L_T$ is often taken to be some average of displacement scales over a

given depth bin. However, this fails to recognise that the enclosed nature of an overturn can set a natural envelope to the estimation (Mater et al., 2015), so that moving through the profile and summing displacements, one can see the start of an overturn and then maintain the sum of displacements until the nett displacement is brought back to zero (within some error). This has the same effect as the centred length-scale proposed by Imberger and Boahash (1986) whereby displacements were aggregated at the centre of the overturn. By using the microstructure temperature sensor record, the lower limit to this scale

is has a smaller spatial resolution than a traditional CTD thermistor sensor.

## 3    Results

The nature of the high flow rates in the strait is illustrated with a day-long sub-section of the two years of velocity data from the eastern side of the Strait (Figure 4). The relatively poor data depth coverage is due to instrument tilt, which while remaining within usable tolerances, does exacerbate side-lobe interference from the surface. While predominantly north-south, the vector sum indicates local speeds reaching 3 m s$^{-1}$ at a water depth of around 30 m (speeds above are not known).

The flow at this location is not symmetric, with southward flows being 20-50 % smaller. Vertical velocities reach 0.1 m s$^{-1}$ with greater high frequency variability when compared to the horizontal flow signal – this compares with variation observed in profiler drop speed (Figure 3). Backscatter structure has some correlation to the flow speed, with the fast flow periods heralding increased backscatter through most of the measured water column. The bulk velocity shear is described in Stevens (2014) and the asymmetry is particularly clear with levels reaching maximum values of +/- 0.01 s$^{-1}$.

The comparison of moored and remotely sensed data (Figure 5**Error! Reference source not found.**) suggests that, despite the energetic nature of the strait, it is not fully mixed during the austral summer (Stevens, 2014). The data are insufficient to indicate if the strait is often stratified in density but it is clearly not homogeneous in temperature for a significant portion of the year. Temperature differences between bed and surface are as large as 3 $^{\circ}$C (primarily in the November-April period). Considering the same data in T-S space (Figure 6) shows the seabed and surface temperatures span the same range

essentially. Three selected microstructure profiles (A, B and C) demonstrate the vertical structure with vertical density differences reaching as high as 0.5 kg m$^{-3}$ over the full depth of the water column. The low salinity data (S<~ 34.4) is seen in Stevens (2014) and results in a 5-month long period at the start of 2012 where the eastern mooring sustained lower S, but kept a similar T to other moorings at the time. The profiles come from right at the end of this period and so do not exhibit anomalous salinities.

Before considering the turbulence data en masse, it is useful to look at some details of selected profiles. The example profile A (Figure 6) is one of the more strongly stratified observed in the strait. The details of this profile (Figure 7) illustrate the effect of the conductivity sensor being un-pumped. However, the profile structure at the macroscale is monotonic in temperature and so temperature displacements are dynamically meaningful. Stratification persisted throughout the water column with N$^2$ being around 10$^{-5}$ s$^{-2}$. Neither the N$^2$, nor the dissipation rate structure varied greatly through the water column.

Near-surface values of $\varepsilon$ were low, but increased to hold a near-constant level through most of the water column, then rising near the bed. The large central overturn, as identified with the Thorpe analysis, contained the majority of the vertical variability in $\varepsilon$ in the profile supporting the decision to keep L$_T$ and $\varepsilon$ calculations separate. The diffusivity proxy (panel c) is notable that in this one instance, the combined $\varepsilon$ and N$^2$ imply K$_z$ exceeds 0.1 m$^2$ s$^{-1}$, i.e. very large. As will be returned to in the Discussion, Bluteau et al. (2017) find that these large mixing events might themselves be underestimated.

The profile B (Figure 8) differs from profile A in that it has a large quasi-homogeneous upper portion of the water column. Stratification results in a reduced N$^2$ being as low as 10$^{-7}$ s$^{-2}$ but increasing with depth. The dissipation rate structure increased with depth through the water column (i.e. in tandem with the stratification). The weak stratification was still

sufficient that overturn scales were small throughout the water column except for the large upper overturn that exceeds 80 m in scale. Interestingly this coincided with an upper layer of *low* dissipation rate. However, this may be due to a thin low salinity surface layer (c.f. Bowman et al. 1983) with a compensating low temperature, and is a case where density rather than temperature should be used to gauge overturns. In this example, the combined $\varepsilon$ and $N^2$ imply a $K_z$ proxy peaking at around
0.1 $m^2s^{-1}$ but mostly an order of magnitude smaller.

        The final profile example described here, profile C (Figure 9), sustains a lower quasi-homogeneous region of the water column. Stratification results in $N^2$ having a baseline around $10^{-6}$ $s^{-2}$ but significantly increasing at the interface zones. The dissipation rate structure here is bi-modal with a mid-depth minimum. Overturn scales followed the dissipation rate trend with an especially large structure near the bed. Dissipation rates at the bed exceeded $5x10^{-6}$ W $kg^{-1}$. The variability in $\varepsilon$
dominates that of the $N^2$, so that the $K_z$ proxy structure mirrors $\varepsilon$ closely, peaking just under 1 $m^2s^{-1}$ near the bed. This extremely high value is to be expected in a flow known to move large boulders.

## 4      Discussion

**Are the dissipation rates actually large?**

15        The distribution of dissipation rate (Figure 10a) shows the level of turbulent kinetic energy (as inferred by $\varepsilon$) extends over five orders of magnitude. While the linear average is around $2x10^{-6}$ W $kg^{-1}$, extrema can exceed $10^{-4}$ W $kg^{-1}$. In addition, most unusually, there were almost no estimates down at the instrument noise floor around $10^{-10}$ W $kg^{-1}$. Scaling these estimates over depth, taking the perspective of a numerical modeller looking to resolve friction losses through a Strait, suggests between 0.6 and 30 W $m^{-2}$ are lost through turbulent dissipation (c.f. say Bab el Mandab of a maximum around 0.2 W $m^{-2}$, Jarosz et al.
20    2011).

        It is easy to ignore bulk Re in ocean physics, assuming correctly that any Re calculation will be "large". However, at the turbulence scales buoyancy can potentially affect overturns and re-stratification. The turbulent buoyancy Reynolds number $Re_b$ ($=\varepsilon/[\nu N^2]$) identifies how velocity fluctuations, and any associated buoyancy flux, evolves and decay. In the present Cook Strait data, the majority of $Re_b$ estimates exceed 100, with the peak of the distribution being around $5x10^4$ (Figure
10b**Error! Reference source not found.**) confirming that the turbulence is "energetic" (Mater et al., 2013). The larger $Re_b$ values exceed $10^7$, which is primarily due to the small $N$ which approaches the levels of detection. This is larger than the range observed by Wesson and Gregg (1994) who, in the much stronger stratification of Gibraltar, saw $Re_b$ values more commonly around $10^2$-$10^3$, but still with some $Re_b$ reaching $10^5$ or more.

        It is a particularly challenging environment to profile in, due to the fast flows and strong winds, combined with the
relatively long profile durations. A profile and retrieval pair would take around 30 minutes to complete, in which time the vessel would have shifted as much as several km. Keeping the vessel on station was not possible as the instrument line would

pay out so great a distance that line-drag would mean that free-fall would cease. Moving the vessel with the line proved too risky in terms of entanglement. Consequently, sequences of two to three profiles were recorded before repositioning the vessel.

Other sampling strategies have been considered, both as a comparison and as a way to extend the dataset. Ocean glider-mounted microstructure would be affected by the substantial vertical flows. Bed-mounted turbulence sampling will be subject to mooring blow-down so that the sampling package will be constantly moving through the vertical. Surface-floating gear is affected by the very substantial surface wave field. In the instance of Cook Strait, free-drifting mooring-based sampling is unlikely to get regulatory approval due to the potential for fouling on submarine high voltage DC cables that cross the strait.

Traditional microstructure profiling thus appears to be the most suitable option for now as we seek to capture a greater variety of conditions, especially during the spring tides. The fast flows mean an ability to rapidly reposition is thus an advantage, meaning a smaller vessel in good weather was a better option than a larger vessel able to handle rougher conditions. The end result of all the trade-offs was that we have yet to work out a way to capture a regular sequence of profiles through a tidal cycle in effectively the same location. However, we have built up a dataset through all phases of the tide, though only from a limited set of seasonal conditions and not in the very fastest flows.

**Does the Thorpe Scale vary systematically with the Ozmidov Scale?**

A cross-comparison of $L_T$ with $L_{Oz}$ (Figure 11) shows a systematic co-variation but one that is far from 1:1. In addition, no $L_T$ greater than 100 m were observed despite the water column exceeding three-times this and with weak stratification. The calculated $L_{Oz}$, on the other hand, is not actually physically constrained and in several instances, it exceeds the water depth. Considering $\log_{10}$ distributions of $L_T$ with $L_{Oz}$, the observed Thorpe displacement scale $L_T$ is substantially smaller than the buoyancy-controlled limit $L_{Oz}$, by an order of magnitude at smaller length-scales. The two estimates come closest at around $L_T \sim$ 10m (being around 50% of $L_{Oz}$). Wesson and Gregg's (1994) observations of turbulence quantities in the Strait of Gibraltar found that the $L_{Oz}$ ($L_B$ in their notation) compared essentially 1:1 with $L_T$, with most estimates falling within a factor of 4 either side. They also found this degree of scatter held throughout the water column. This differs from that seen here (Figure 12) where the $L_T$ is substantially smaller than the $L_{Oz}$ by as much as a decade at smaller scales. The scatter is also larger in the present data as this also is around a decade either side of the mean value. This latter point may be driven by the present noise-rejection conditions resulting in fewer very small $L_{Oz}$ (say <0.5 m) whereas the Gibraltar data drop to as low as $10^{-2}$ m. In addition, the present use of the microstructure sensors to estimate $L_T$ allows this to extend to smaller values. Furthermore, the Cook data exhibit a possible split in behaviour around $L_{Oz}=10$ m whereas the Gibraltar data only hints at this. Making the same comparison with the Dunkley et al. (2015) Gulf of Aqaba observations ranging over $L_T=0.1$-10 m, the distribution is almost a mirror reflection around the 1:1 line from that observed in Cook Strait. In the Gulf of Aqaba results, the $L_T$ exceeds on average the $L_{Oz}$ by as much as an order of magnitude – a trend also seen in the Bluteau et al. (2013) data. Finnigan et al. (2002) used the $L_T$ approach to estimate turbulence in the vicinity of a submarine ridge, and cross-

comparison with strain-derived estimates of ε (e.g. Frants et al. 2013) suggested it was applicable at least where there was detectable stratification. However, the dissipation rate levels were around $10^{-9}$ W kg$^{-1}$, three orders of magnitude less than in the present situation. While field studies are typically compromised in some way, complementary analyses through direct numerical simulation (e.g. Smyth et al., 2001) provides supporting evidence that there should be a systematic variation in the empirical overturn scale ($L_T$) and the buoyancy-induced limit to overturns ($L_{Oz}$). This approach suggests that the variation in the ratio of the two scales is an indication of the age of the mixing event, with $L_{Oz}$ increasing relative to $L_T$, and so that scatter in real observations reflects the random age captured by sporadic profiling.

One of the challenges in ocean turbulence is that studies are so intense, focused and idiosyncratic that they tend to be analysed in isolation and rarely synthesized. As a counter-example to this, Mater et al. (2015) collated three open-ocean turbulence experiments from (i) the North Atlantic at around 3000 m (NATRE, Toole et al., 1994), (ii) Brazil Basin mid-Atlantic at around 3000 m (BBTRE, St. Laurent et al., 2001) and (iii) Luzon Straits at around 2500-3000m, (IWISE, Alford et al., 2011). Here we consider the present data in this context (Figure 12). The ratio of $L_T$ to $L_{Oz}$ in these deep-water experiments was considered against a $L_T$ non-dimensionalised by the length-scale extracted from viscosity and buoyancy $(\nu/N)^{1/2}$ representing the distance momentum can diffuse in a time $N^{-1}$. All follow the same trend of the ratio $L_T/L_{Oz}$ growing with increasing eddy size. All but the NATRE data have significant proportion of data lying with $L_T < L_{Oz}$. The present Cook Strait data illustrate this aspect most strongly nearing an order of magnitude smaller at low $L_T$. Furthermore, the present data extend into the largest non-dimensional $L_T$ space. Mater et al. (2015) suggest that while the experiments are deep-water they are still constrained vertically by convective scales.

A comparison of direct shear probe dissipation rate estimates of $L_{Oz}$ and the Thorpe Scale $L_T$, indicates a broadly comparable trend but that the comparison is not 1:1 (Figure 11) with the departure growing for larger scales. There looks to be a bias towards high $L_T$ values for low $L_{Oz}$ value at shallow depths. Using the Dillon (1982) approach of considering the $L_{Oz}=[\varepsilon/N^3]^{1/2}$ and assuming $L_{Oz}/L_T$ is fixed such that $L_{Oz}=aL_T$, then we arrive at a simple expression for ε (Figure 13). This compares the dissipation rates from each $L_T$ overturn with both the direct and $\log_{10}$-based average ε within that overturn. The direct average (squares) provides a close comparison between observed and estimated ε. This agreement holds from $2\times10^{-9}$ W kg$^{-1}$ through to $2\times10^{-5}$ W kg$^{-1}$, with only one or two departures. The most notable being at $10^{-9}$ W kg$^{-1}$ where it is biased high by a very larger outlier that is so anomalous that it should possibly be discounted. There is an obvious family of outliers in the upper 30 m of water that are anomalously high in terms of the parametrised estimate $a^2L_T^2N^3$ of dissipation rate. Most likely this is a result of some surface-driven stratification effect that either (i) affects turbulence in some systematic way, or (ii) confounds the temperature-based density correction. The log-based comparison is around an order of magnitude smaller. This is included in order to compare this representation with Figure 11.

While the $L_T$ never approaches the full water depth, they are large given the flow speeds. Stevens (2014) measured velocity shear at bulk scales (i.e. resolved from 8 m ADCP bins) reaching as high as 0.01 s$^{-1}$. The velocity variation over an

eddy of $L_T=100$ m in a flow with a velocity shear of 0.01 s$^{-1}$ is 1 m s$^{-1}$. This is comparable, but not greater than, background speeds suggesting that it might influence the degree of isotropy by straining eddy structure in the horizontal direction. A similar effect should be expected in slower but much deeper systems such as Bussol Strait (Tanaka et al. 2014; Bryden and Nurser 2003).

**Implications for, and of, mixing rate estimates**

The $\Gamma=0.2$ "constant" is a clear point of contention in the literature (e.g. Dunkley et al., 2012; Bluteau et al., 2013; Mashayek et al., 2013). Bluteau et al. (2017) develops an approach that takes microstructure profiles and resolves the diffusivity "directly" fitting a model for dissipation of thermal variance to the convective-inertial subrange (i.e. lower

wavenumbers than the dissipation scale). The Bluteau et al. (2017) analysis suggests that improved estimation of the thermal diffusivity indicates that the fixed mixing coefficient might underestimate mixing by a factor of 5 in the mean especially for the more turbulent events. Extending this by applying the Osborne diffusivity method sees an average diffusivity is around 0.04 m$^2$ s$^{-1}$ and exceeding 1 m$^2$ s$^{-1}$ (Figure 10b). One might expect a 300 m water column to then be homogenised in a time ($L^2/K_z=$) 300$^2$/1=25 hours, but this might be as little as 5 hours if the Bluteau et al. (2017) increased estimate of $K_z$ were to

hold. Tidal excursions due to the semidiurnal tide are insufficient to flush the strait in a single cycle. Indeed, with a net drift of around 0.02-0.1 m s$^{-1}$ (Stevens, 2014) it takes many tidal cycles. This suggests that, at these most energetic of mixing conditions, we should not expect to see a stratified water column as it should get mixed over the multiple tidal cycles it takes for water to clear the strait. The bulk top-bottom observations (Figure 5**Error! Reference source not found.**) counter this as, for some of the year at least, there is clearly a scalar gradient. Possibly, the observations need to be restructured and collected

drifting with the flow to better follow the evolution of mixing.

Lafuente et al. (2013), in their exploration of the impact of vertical diffusion of biologically relevant scalars in the Straits of Gibraltar, found a highly two-dimensional situation whereby the mixing is highly spatially variable, with the presence and location of an internal hydraulic jump being very important. In a similar way to Cook Strait, their simulations show, despite the reasonable tides and strong estuarine circulation, it takes some time for well-mixed water to exit the system.

Lafuente et al. (2013) set their background vertical diffusivity to 10$^{-7}$ m$^2$ s$^{-1}$ and also prescribed a maximum of 10$^{-2}$ m$^2$ s$^{-1}$ in order to "avoid unrealistically high values". While having the potentially very small $N^2$ in the denominator for $K_z$ is problematic, the very large $\varepsilon$ and $L_T$ make it reasonable to assume, with finite N, that the larger $K_z$ estimates are useful in a bulk sense. This suggests future work could apply the approach of Bluteau et al. (2017) to profile data to capture the large $K_z$ events.

While the focus here is on vertical structure and mixing, the horizontal perspective is also of value. The Strait has been identified as a dividing line in terms of ecological structure (e.g. Forrest et al., 2009). The implication is that there is not a great deal of transverse (across-strait) transport. This supports the focus of the present work on the vertical structure.

Furthermore, over the time it takes to drift through the strait all vessel tracks tended to be on an axis aligned with the strait. Over these scales of time and space the strait itself is bathymetrically reasonable consistent. It remains to conduct a study that will adequately quantify across-strait mixing, the associated drivers and the moderating influence of vertical mixing.

## 5 How does the turbulence compare with other straits?

While the present focus is on turbulent length-scales rather than their oceanographic context. Studies examining flows through stratified straits, both in a net sense and in exchange conditions, classically view the mechanics in terms of non-mixing internal hydraulics (Helfrich, 1995; Hogg et al., 2001). This enables identification of phenomena such as control points and the presence of hydraulic jumps. The extension to consider the role of turbulence and mixing in influencing the system

uses bulk estimates of $K_z$ (Hogg et al., 2001). They were able to demonstrate that by varying the mixing coefficient a strait system could vary between inviscid hydraulic conditions through to a mixing layer. This highlights the need for more direct observations of mixing in such situations.

While Stewart and Grant (1999) identify the high Reynolds number in Seymour Narrows (Discovery Passage, British Columbia), it is clear that deeper costal systems like Cook Strait and much deeper oceanic constrictions (e.g. Tanaka et al.,

2014) create even higher Re conditions. It is difficult to draw general conclusions describing strait behaviour from any one situation as Gregg and Özsoy (2002) noted when quoting Tolstoy to highlight field idiosyncrasies. While the quote was in the context of the Bosphorus, the canonical strait at this scale is probably Gibraltar, the scene of some of the first systematic turbulence quantification (Wesson and Gregg, 1994). These authors state that their 1994 results "rather than being definitive, these results are only the beginning of turbulence measurements in the Strait of Gibraltar". While this has not really turned

out to be the case for Gibraltar, the approach and results spawned a range of studies in comparable systems (Table 1), with the ensemble providing a natural laboratory for exploring a range of ocean mixing phenomena.

**Acknowledgements**

The author would like to acknowledge colleagues who have aided in the work, in particular Brett Grant, Mark Hadfield, Fiona Elliott, Craig Stewart, Ross Vennell, Murray Smith, Steve Chiswell, Graham Rickard, Rebecca McPherson and Joe O'Callaghan. Two anonymous Reviewers are thanked for providing constructive comments on an earlier version of this manuscript and Peter Baines is acknowledged for identifying the Tolstoy quote as coming from Anna Kerenina. The support of NIWA Core Funding, the New Zealand Sustainable Seas National Science Challenge and the Royal Society Te Apārangi

Marsden Fund is acknowledged.

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

Table 1 Comparison of approximate representative strait scales (extended from Helfrich, 1995 and Hogg et al. 2001)

| Strait | $g'$ m s$^{-2}$ | Depth m | U m s$^{-1}$ | Length km | Re | Ri | Source |
|---|---|---|---|---|---|---|---|
| Cook | 0.006 | 350 | 3.0 | 40 | $10^9$ | 0.9 | Present study; Stevens (2014) |
| Bosphorus | 0.12 | 35 | 0.8 | 30 | $3 \times 10^7$ | 2.1 | Gregg and Özsoy (2002) |
| Cordova | 0.003 | 30 | 0.9 | 3 | $3 \times 10^7$ | | Lu et al. (2000) |
| Seymour Narrows | - | 60 | 6 | 3 | $3 \times 10^8$ | | Stewart and Grant 1999; Lueck et al. (2002) |
| Gibraltar | 0.02 | 280 | 1.2 | 20 | $4 \times 10^8$ | 3.6 | Wesson and Gregg (1994) |
| Bussol | 0.01 | 1750 | 1.0 | ~50 | $10^9$ | | Tanaka et al (2014) |

**Figures**

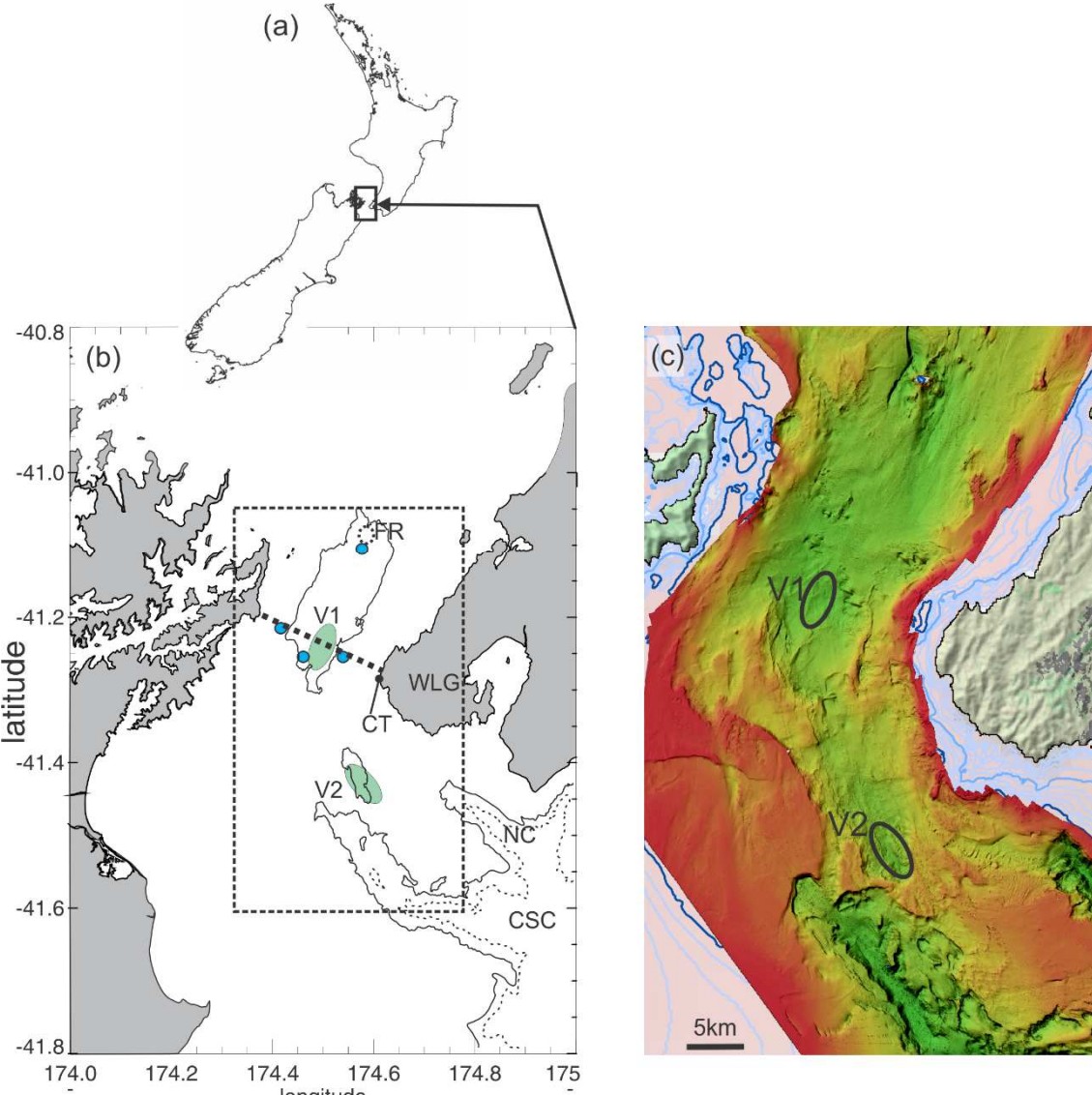

Figure 1 Location showing (a) New Zealand, and within this, (b) Cook Strait Narrows is bounded by Cape Terawhiti (CT) to the east and the headlands of the (shaded) Marlborough Sounds to the west and with the Cook Strait and Nicholson Canyons to the south (CSC and NC).

5   The 200 m (solid) and 400 m (dashed) depth contours are marked, as well as the shoal at Fishermans Rock (FR). ADCP moorings are marked with blue circles. The microstructure data come from profile regions V1 and V2.

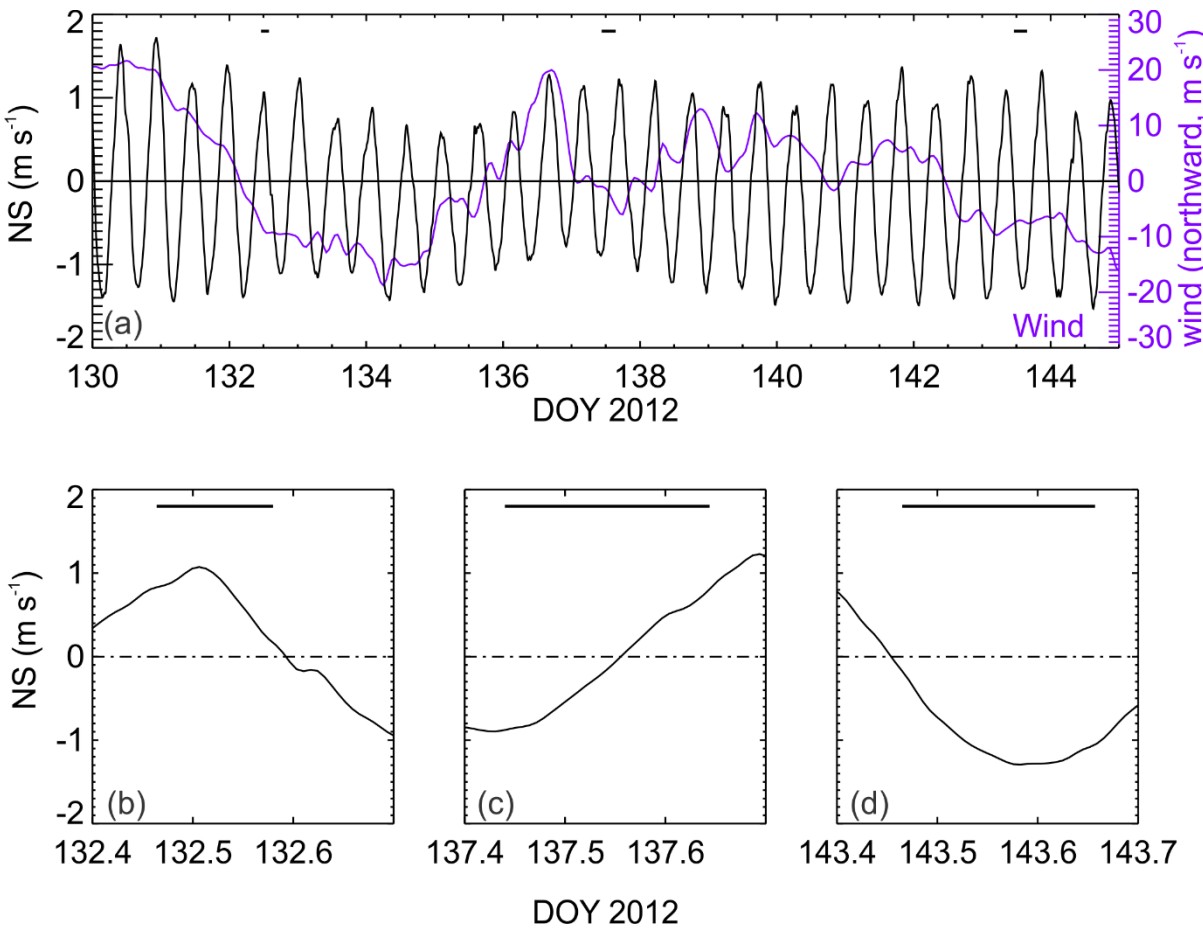

Figure 2 Sampling conditions showing average N-S water column velocity between 60 and 100 m depth and wind speed - both filtered with an hourly low-pass filter. The bars show microstructure sample periods. These bars are expanded in the daily sampling relative to tidal conditions is shown in (b), (c) and (d).

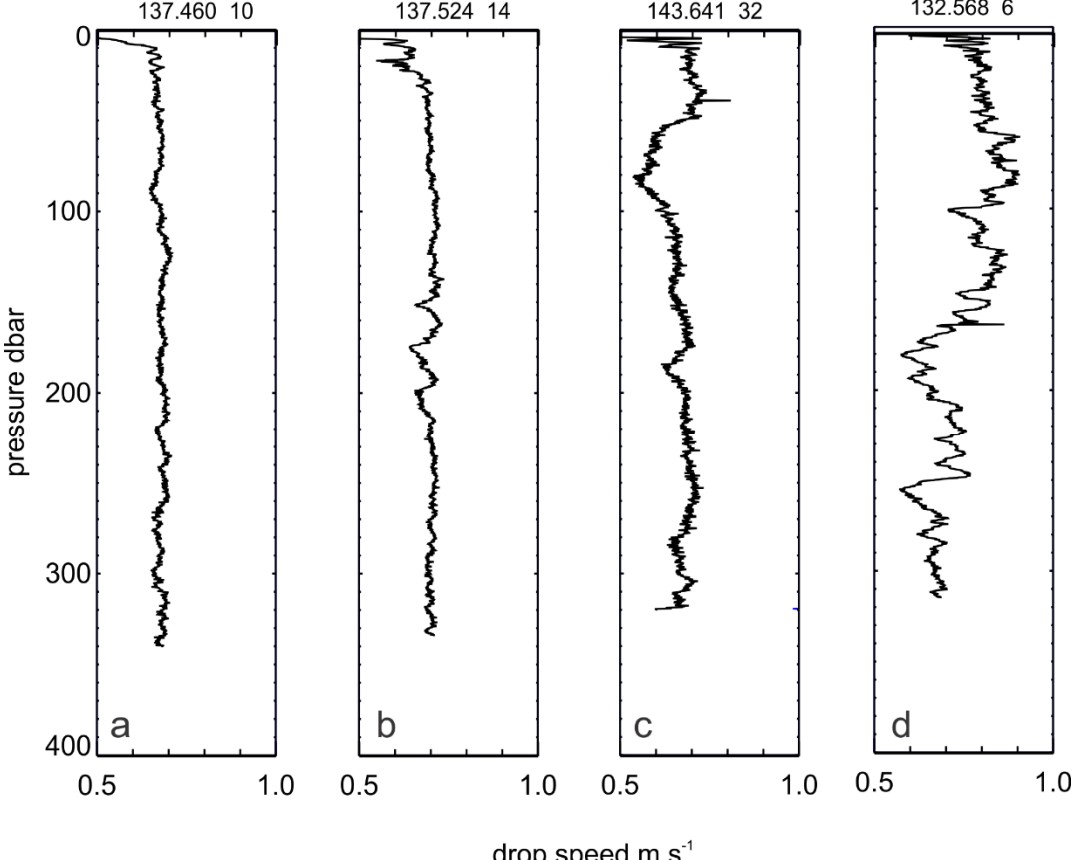

Figure 3 profiler drop speed from a number of example profiles (time and profile number on top).

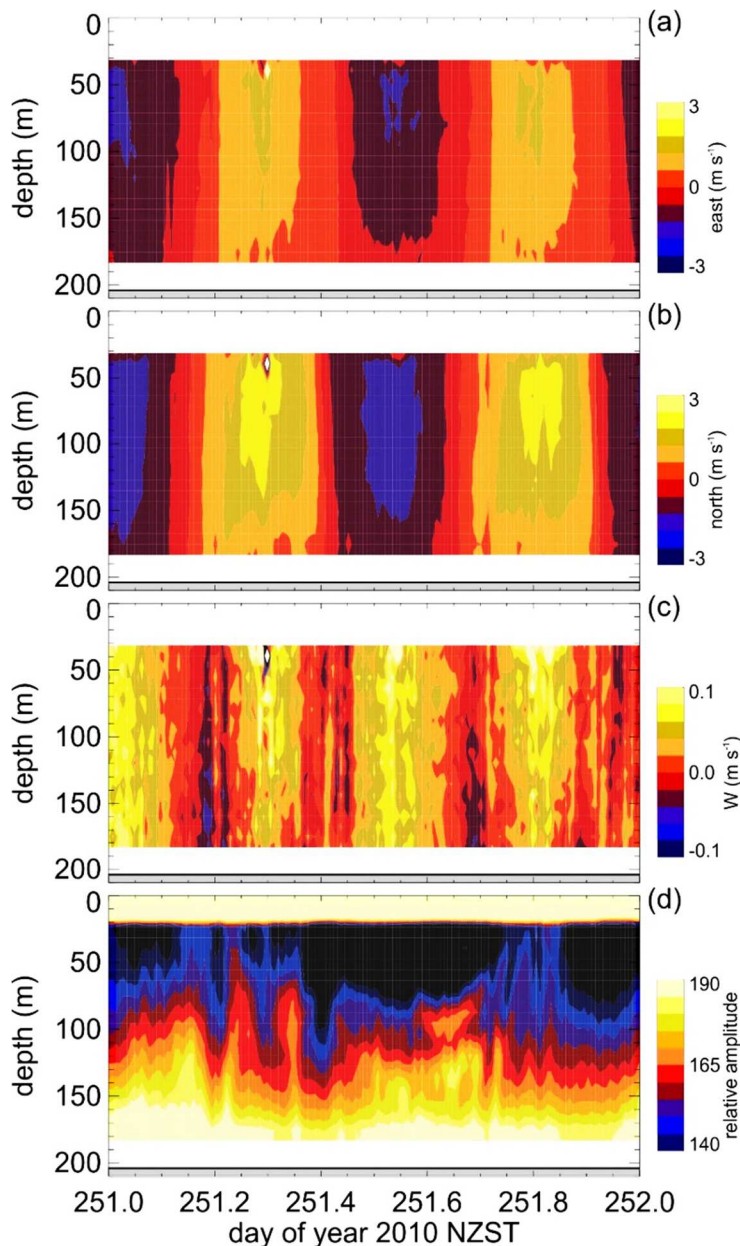

Figure 4 velocity data from eastern side of strait showing (a) east-west, (b) north -south, (c) vertical velocities and (d) backscatter amplitude.

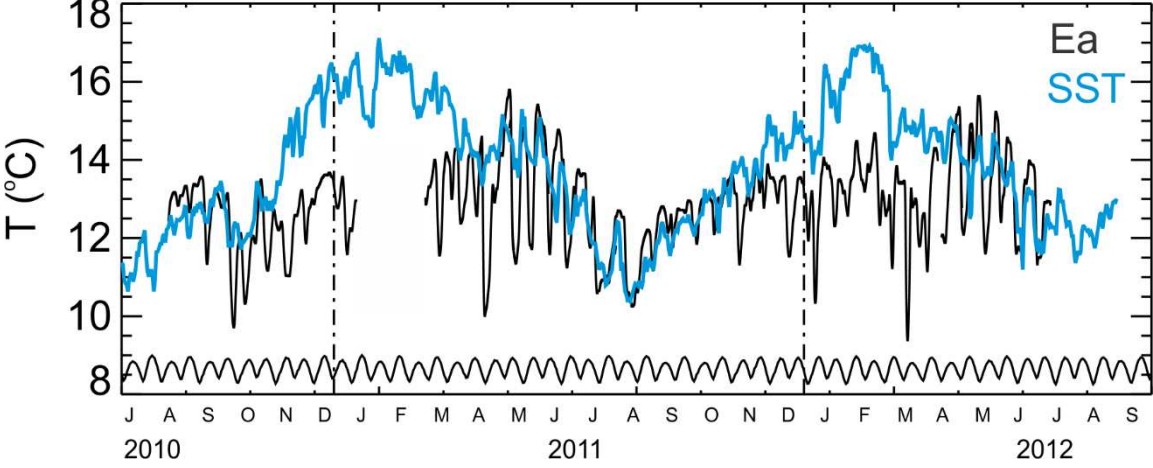

Figure 5 Temperatures from near-bed (Ea, Figure 1) and satellite-derived sea surface temperature (SST). The arbitrarily-scaled spring-neap

envelope is along the base of the panel.

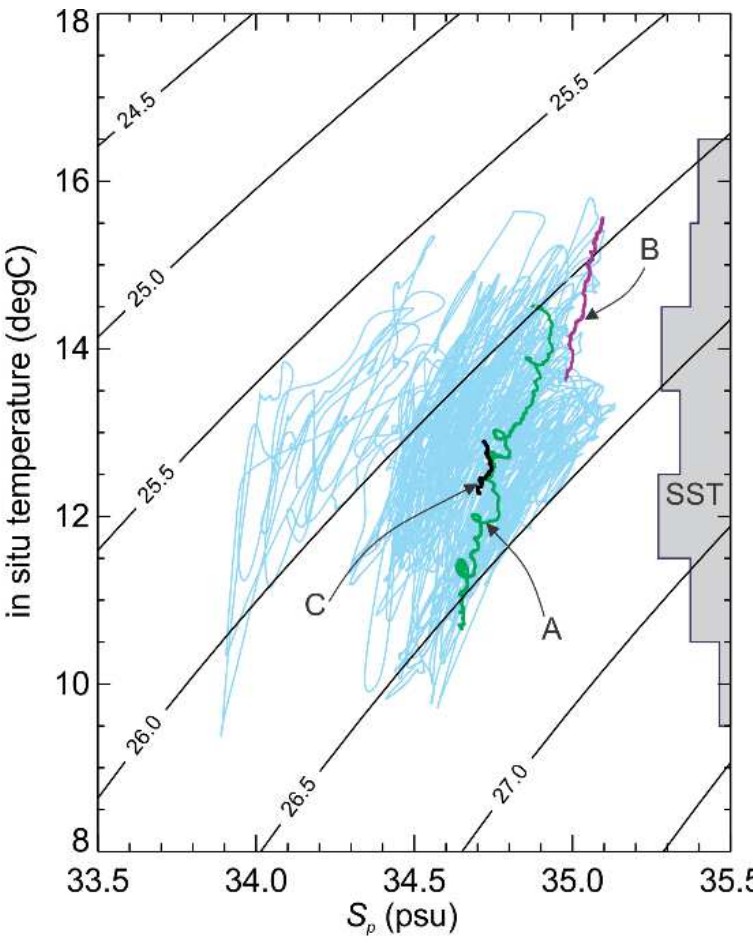

Figure 6 Temperature and salinity ($S_p$) from the seabed sensors on the moorings with profile data overlain (see text for details) and $\sigma_T$

contours. The arbitrary-scaled histogram on the right-hand margin shows the distribution of satellite-derived sea surface temperature (SST).

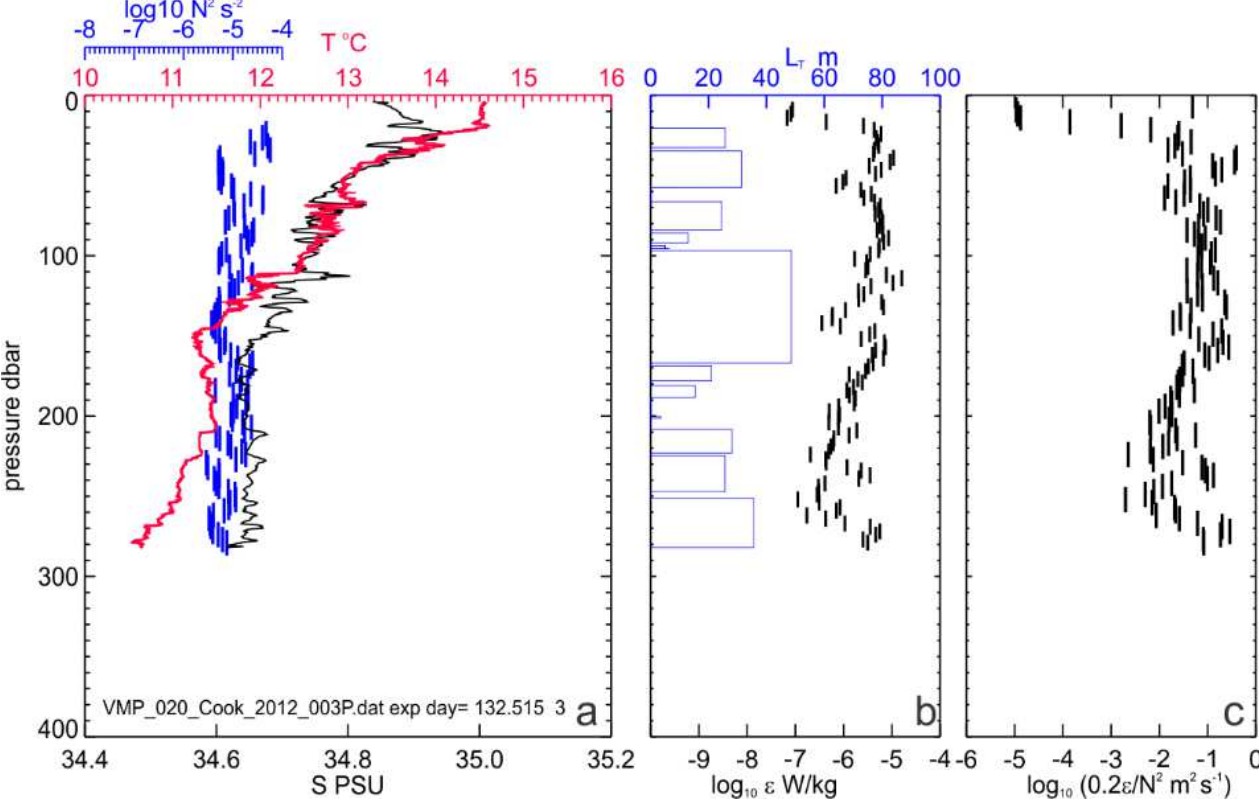

5    Figure 7 Profile A from day 132, 2012 showing (a) temperature, salinity and buoyancy frequency squared ($N^2$), (b) $L_T$ and $\varepsilon$

and (c) proxy for vertical diffusivity $0.2\varepsilon/N^2$. Note the profile extends over the full depth of the water column.

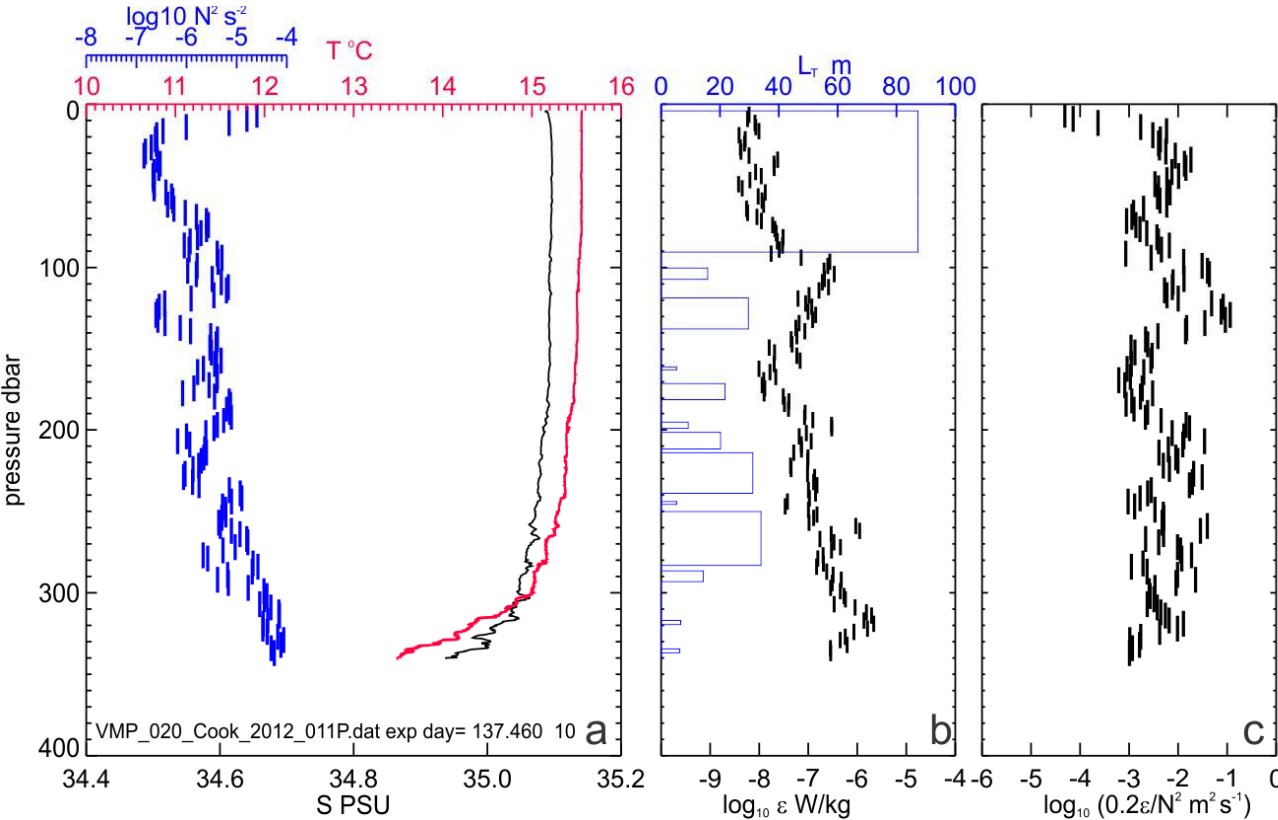

Figure 8 Profile B from day 137, 2012 showing (a) temperature, salinity and buoyancy frequency squared ($N^2$), (b) $L_T$ and $\varepsilon$

and (c) proxy for vertical diffusivity $0.2\varepsilon/N^2$. Note the profile extends over the full depth of the water column.

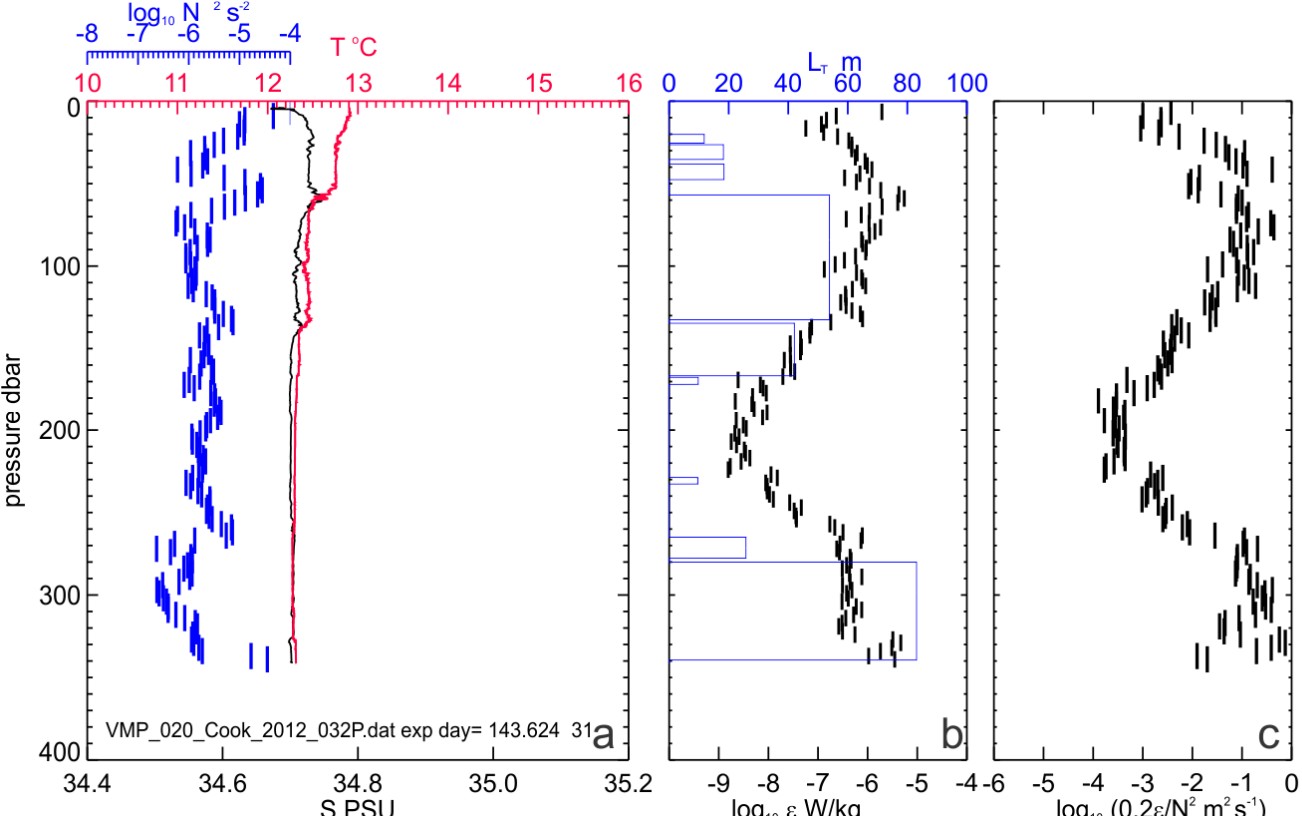

Figure 9 Profile C from day 143, 2012 showing (a) temperature, salinity and buoyancy frequency squared ($N^2$), (b) $L_T$ and $\varepsilon$ and (c) proxy for vertical diffusivity $0.2\varepsilon/N^2$. Note the profile extends over the full depth of the water column.

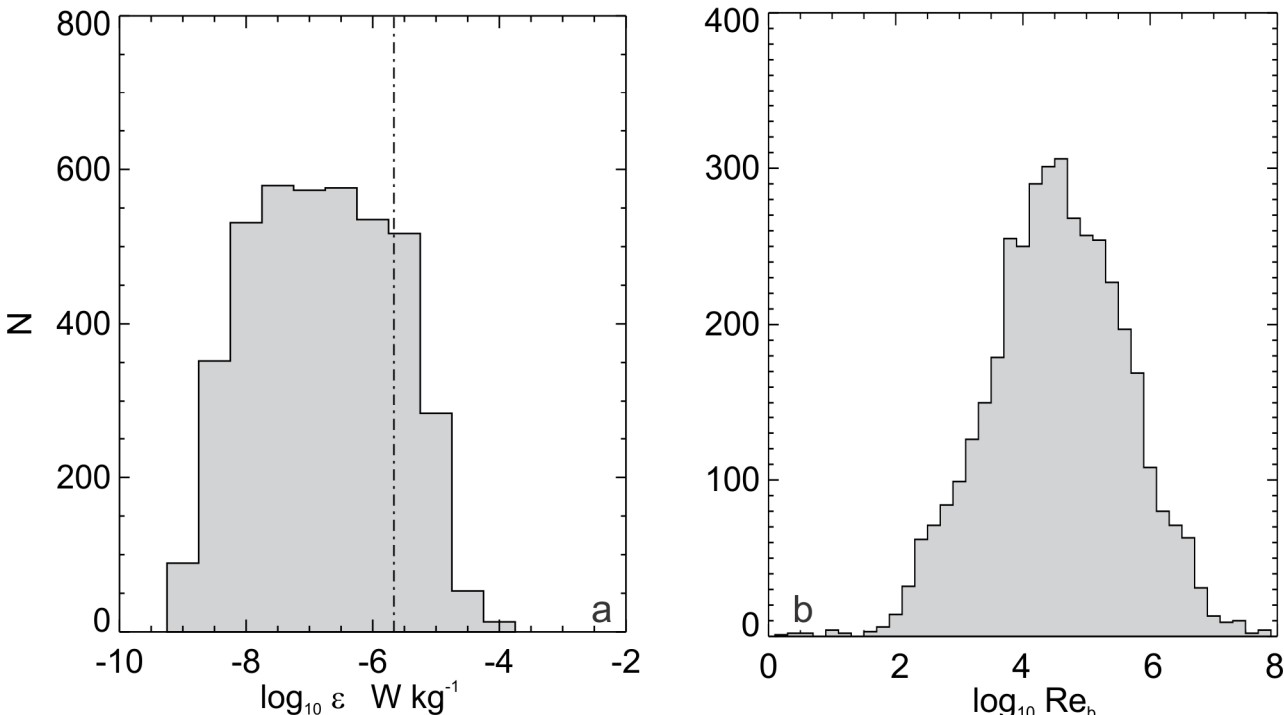

Figure 10 Distributions of (a) dissipation rate ε and (b) buoyancy Reynolds number, Re_b.

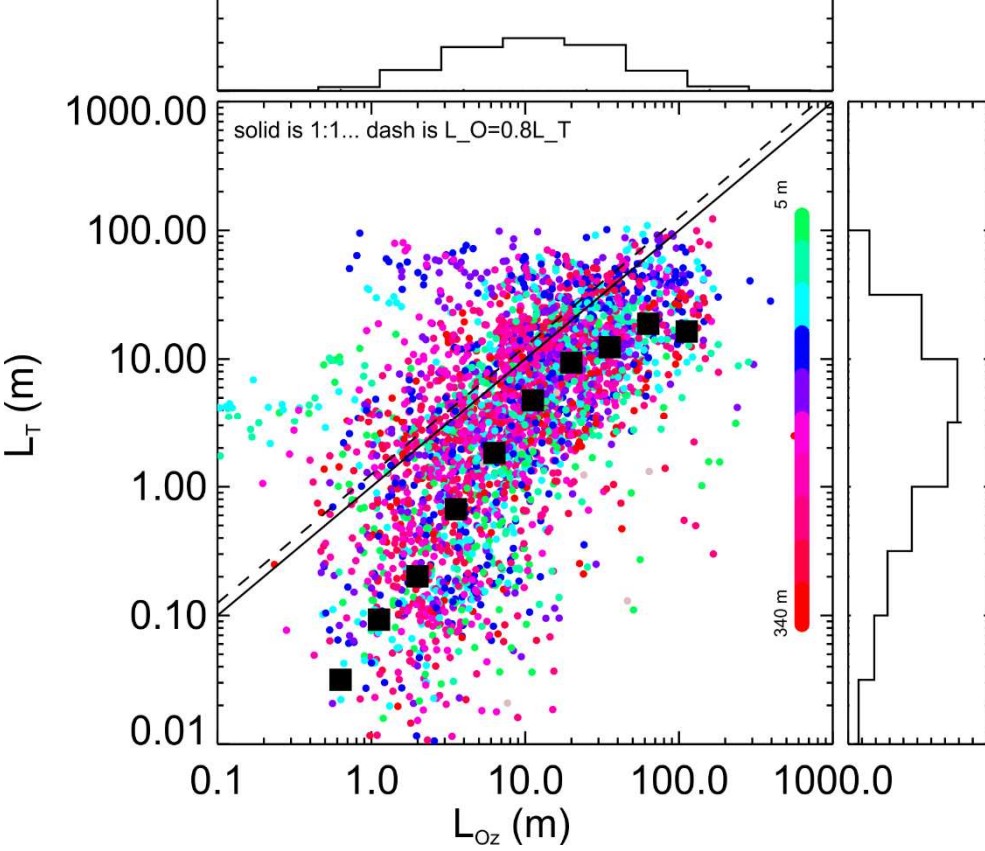

Figure 11 Scatter plot of $L_T$ vs $L_{Oz}$ colour-coded by depth. Lines for unity and for $L_{Oz}=0.8L_T$ are shown and associated histograms of length-scales are shown also. Averages were calculated in log10 space and 0.5 m was considered a lower-bound for $L_{Oz}$.

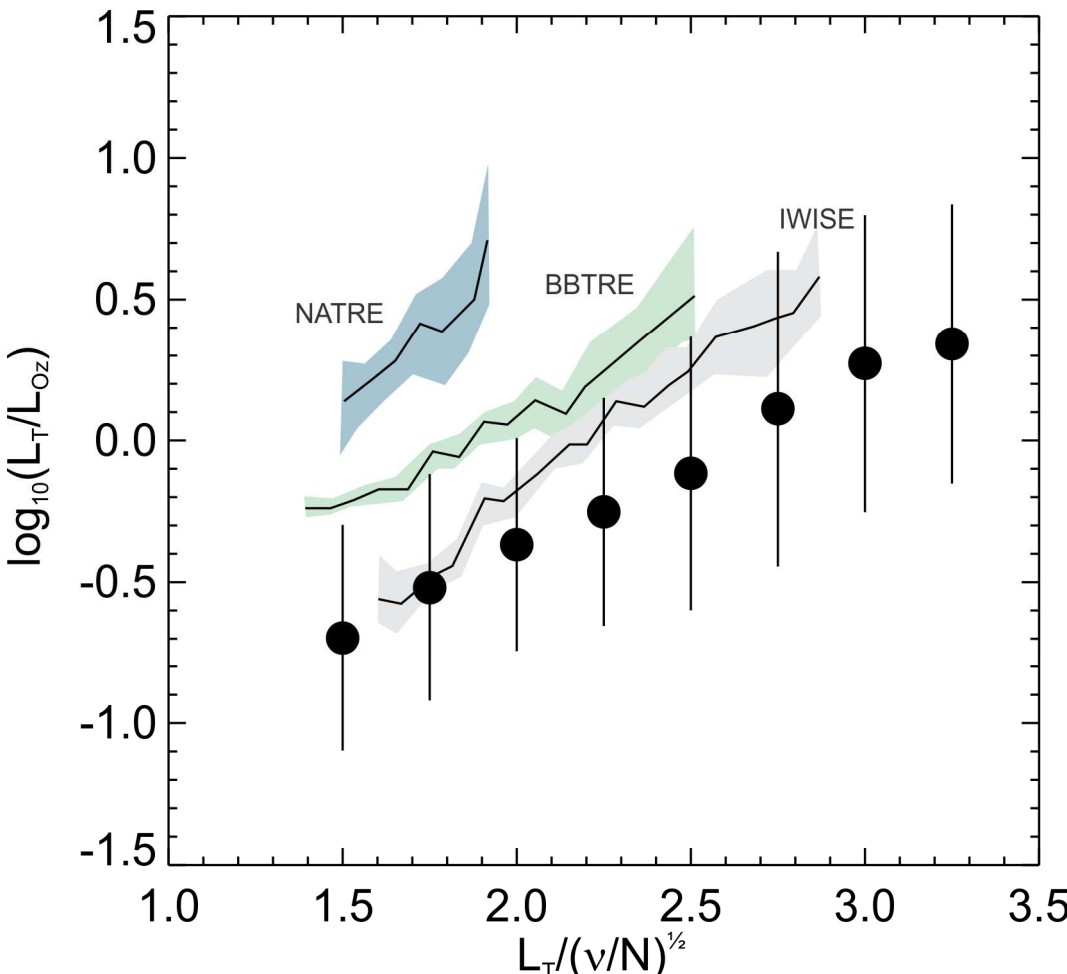

Figure 12 The comparison of the $L_T/L_{Oz}$ ratio as a function on non-dimensionlised $L_T$. The present data set (circles with +/- 1x standard deviation as error bars) is superposed on top of synthesized results following Mater et al (2015).

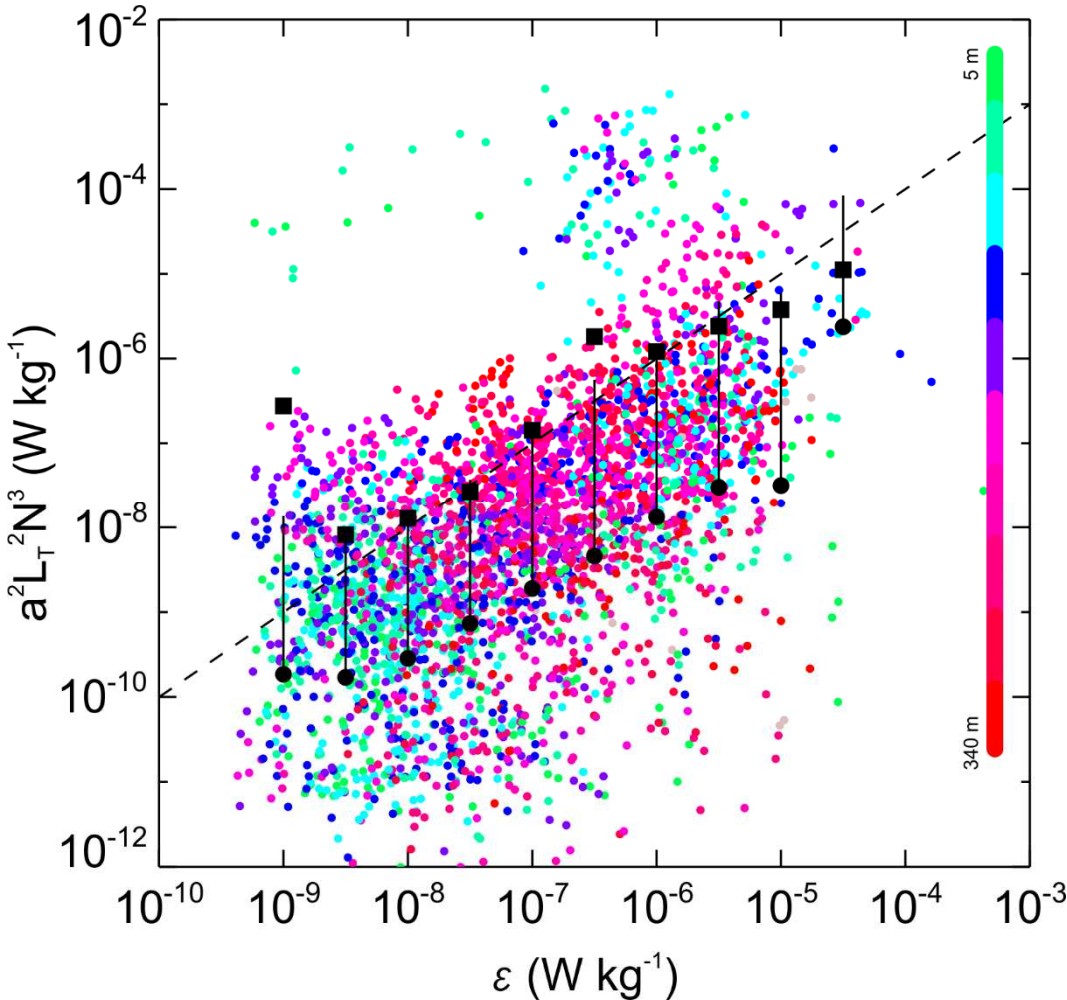

Figure 13 Comparison of dissipation resolved from the $L_{Oz}$ with the direct measure of $\varepsilon$. Averages were calculated in log10 space for $\varepsilon$ and all length-scale-based estimates in that bin were collated. Here the average and average+ 1 standard deviation are shown with a circle-line pair and the average in log10 space is shown as a square. The averages excluded outliers in the surface water as described in the text.

