# Peer review of "Turbulent Length-scales in a Fast-flowing, Weakly Stratified, Strait: Cook Strait, New Zealand"

_Ocean Science, 2017_

## Referee Comment (RC1) · Anonymous Referee #1 · 16 Nov 2017

The comment was uploaded in the form of a supplement:
https://www.ocean-sci-discuss.net/os-2017-30/os-2017-30-RC1-supplement.pdf

---

## Referee Comment (RC2) · Anonymous Referee #2 · 26 Feb 2018

The manuscript discuss direct measurements of turbulent quantities in Cook Strait. Considering that such data are relatively scarce in oceanography and can be interpreted in a broader context, the reported data are valuable.

The central issue of this manuscripts and its main message is about the comparison between the Thorpe and Ozmidov scales. Unfortunately, the discussion itself is rather short and poorly documented (three references). More efforts should be put in the analysis around figures 12, 13 and 14. The manuscript will be greatly improved by a better focus on this scientific issue. In particular, can the data shed new light on the claim (Mater, 2013) that $L_T \approx L_O$ with $Nk \approx \epsilon$ ?

The measurements themselves are presented with (too) many figures, but basic information is missing. Little is said, for instance, on the timing of the collected 34 profiles

covering a very large period of time of 5 years. Processing of the microstructure data must be described or documented in a much more precise way than with sentences like "in the usual way" or "An approach".

Considering the fast flows in this region and the irregular topography, three-dimensional effects (e.g. horizontal advection) are anticipated and should be discussed.

Some of the (many) typos and formal problems to be fixed.

- Page 3, line 4 : "velocity *Sh*" → velocity shear ?

- Page 3, line 8 : "Do we actually observe high dissipation rates?"

- Bottom of page 5 and first paragraph of page 6 : please fill the gaps "xxx" and "X" "Y".

- Please carefully check references : some of them are missing or unused (Gregg and Oszoy, 2002; Matter et al, 2003 or 2005;...)

- Figure 6 seems not to be cited / discussed in the text.

- Many figures are provided but with very little discussion. The ratio *number of figures* to *Length of discussion* seems to be rather low.

- ...

In conclusion, the data are interesting but the manuscript should be better focused to avoid wild discussions of many details (e.g. on individual profiles taken at different (unknown) location and times) and come to solid conclusions.

---

## Author Comment (AC1) · 9 May 2018

see supplementary file for reponse

Please also note the supplement to this comment:
https://www.ocean-sci-discuss.net/os-2017-30/os-2017-30-AC1-supplement.pdf

———————————————————

---

## Author Comment (AC2) · 9 May 2018

please see supplementary file for response

Please also note the supplement to this comment:
https://www.ocean-sci-discuss.net/os-2017-30/os-2017-30-AC2-supplement.pdf
* * *

---

## Author Response (AR1)

11may2018

Dear Editor:  With respect to:

Title: Turbulent Length Scales in a Fast-flowing, Weakly Stratified, Strait: Cook Strait, New Zealand
Author(s): Craig L. Stevens
MS No.: os-2017-30

Please Find the Following:

- Response to Reviewer 1
- Response to Reviewer 2
- Tracked Changes Version of Manuscript

I believe this work documents rarely recorded measurements from a novel location, yielding new insight into aspects of mixing in the ocean with general applicability beyond the sample location.  I thank you and the Reviewers for the time taken to improve the manuscript.

Best regards

Craig Stevens

**Response to Reviewer 1 comments on Stevens (2017) Turbulent length scales in a fast-flowing, weakly-stratified Strait: Cook Strait, New Zealand. (original reviewer comments in black).**

This is an interesting paper reporting measurements of turbulence in a very energetic flow through ocean straits, in this case Cook Strait N.Z. Such measurements are relatively rare and this therefore represents an interesting addition to the literature on direct measurements of ocean turbulence in energetic flow. That said, the paper is poorly presented with some important details about the measurements not included in the paper, and even some typographical errors. These issues need to be addressed before the paper is suitable for publication.

*I thank the Reviewer for their very helpful and clearly knowledgeable comments and suggestions. The recent emergence of multiple papers in the literature on aspects of this topic is further evidence that this research theme is important and widely applicable. The following responds to their points. The lack of inclusion of measurement details is responded to below and describes the associated modifications to the revised manuscript. With regard to the point about poor presentation, a number of typos have been cleaned up. As well as this, their comments have motivated a substantial number of improvements.*

**Detailed comments:**

- P1,L24 Waterhouse et al 2014  - *thanks, corrected.*

- P1,l27 Wesson and Gregg (1994) report measurements in Straits of Gibralter, so why is this "… (a) coastal environment". Koch-Larrouy et al (2015) (DSR, 106:136-153) is also relevant here. – *fair enough, I think the point of difference relates to what is a coastal environment and the mechanics of "influence". The initial reference was to bring attention to the effect of strong tidal mixing but I am happy with the reviewer's suggestion as well and additional reference that connects to high trophic levels (Scott et al. 2010) and have modified the text accordingly.*

- P2, L20 . Energy bearing scale. Why is *LT contained* by *LO* , they are independent lenghtscales? *This is of course a key question to ask and is at the heart of the study and many others. The Reviewer asks in what sense are they independent? LT is empirical and $L_{Oz}$ is a scaling argument - but of the same mechanics. What about "constrained by the $L_{Oz}$"? The relationship between LT and $L_{Oz}$ is key to the manuscript and many papers that seek to quantify dissipation rate from overturn scale. I have changed to "constrained" and return to this dependence in the Discussion – "The calculated $L_{Oz}$, on the other hand, is not actually physically constrained and in several instances it exceeds the water depth".*

- P5, L18- "The microstructure data were processed in the usual ways resolving the dissipation" is insufficient. Is the author speaking of using the Naysmith empirical spectrum? More detail is needed here. Bluteau et al (2017, JTECH, 34: 2283-2293) provides an extensive review of processing methods for free-fall profilers, and also provides insight into how to process fast-response temperature measurements, and it may well be possible to apply these ideas here. See below. *Reviewer Two highlights this also. This is a debatable point as the field has evolved that there is now a consistent set of hardware and data processing available. For example, the canonical Wesson and Gregg 1994 paper addresses such*

*points of clarity and so the numerous studies in the intervening quarter of a century fill in many of these issues. For example, Bluteau et al. 2017 refers to an earlier paper for shear microstructure methods. On reflection, I should not have used the phrase "in the usual way". The Reviewer is correct in that there are always points of clarity and interest in following through on these aspects. In acceding to the Reviewer's point, I now include the reference to Bluteau et al. (2017) which was published after the initial submission of the present manuscript and include additional information regarding the processing. I do note that the original manuscript included a figure and discussion of variability in drop speed which is rarely discussed in available studies. I have made this contribution clearer in the revision.*

- P5, l23 what is xxx? *Thanks for spotting this as there were some version control issues. This returns to the point above about the relationship between LT and LO. The amended text now says "One might expect overturns, as identified using the LT, to be equal to, or smaller than LOz. Dillon (1982) observed the ratio to be LT/LOz =0.8. This calculation struggles with regions of weak stratification where locally-small N2 drives a very large scale. This makes sense as weak stratification fails to retard turbulence. However, it can lead to non-physical outcomes as the scale will eventually exceed water depth."*

- P6, L4 Ranges of Γ are missing. See Bluteau et al (2017) and references therein. *Thanks. The ranges weren't missing, they were not specified. This is helpful as the Bluteau reference was not available at the time of writing the initial manuscript. Although to be fair this reference doesn't clearly identify the ranges specifically – it's concluding remarks say "The estimated Rif varied over almost two orders of magnitude with a median Rif not significantly different from the canonical value of 0.17. The median Rif obtained from either technique did not differ significantly from this value, although the median Rif obtained from the fitted chi estimates were slightly larger the median Rif obtained from the integrated chi estimates". The text has now been amended in a number of places to highlight the results of Bluteau et al (2017).*

- P7, L5 The fact that the Strait is not well mixed suggests that the vertical diffusion time scale $H2$ $Kz$ is long compared to advection times in the Strait? Assuming here that advection is re-establishing the vertical gradient? This is discussed later in paper, but argument is confusing. *The point is important because, in an applied sense, this is a key aspect of the location and experiment. A number of references assume because it is fast flowing and clearly turbulent that it homogenizes the water column. This is not supported by the observations. These observations come from the strait narrows and so presumably represent the most energetic conditions. The Reviewer suggests that it is restratification due to advection. This is possible, but given the spatial heterogeneity and relatively fast transit time it is also possible that the water column simply doesn't have time to homogenize as suggested by the scaling in the discussion. This point is now clarified in both the Results and Discussion which has been amended to say "This suggests that, at these most energetic of mixing conditions, we should not expect to see a stratified water column as it should get mixed over the multiple tidal cycles it takes for water to clear the strait. The bulk top-bottom observations Fig 5) counter this as, for some of the year at least, there is clearly a scalar gradient. Possibly, the observations need to be restructured and collected drifting with the flow to better follow the evolution of mixing."*

- P7, l20. The usual argument is the dissipation rate is dependent on the intensity of the background shear S. Why is it dependent on N? *The N is used here to delineate layers in the water column (at least on a profile-by-profile basis). The text originally was designed to indicate that the dissipation rate and stratification structure were consistent. I agree with the Reviewer that this terminology could be misleading and have reworked the text to not imply direct causality.*

- P7, l23 One has to wonder how meaningful is the calculation of the Thorpe scale $LT$ in this situation. It is a strongly advective situation and vertical stratification is (relatively) weak, so how do these effects conspire here? Some estimates of accuracy of $LT$ scale calculations would be useful, particularly as here we find the scales are large compared to the total depth? *This is an excellent point and one that has been explored in the wider analysis of the problem but not included in the initial manuscript. I believe the Reviewer is getting at the issue that such a large overturn will have time to be affected by the background flow. It is not clear to me that it affects the "accuracy" of the LT but rather it affects what the LT actually means. This is now conside*red in the Discussion which says ..."*While the LT never approaches the full water depth, they are large given the flow speeds. Stevens (2014) measured velocity shear at bulk scales (i.e. resolved from 8 m ADCP bins) reaching as high as 0.01 s-1. The velocity variation over an eddy of LT=100 m in a flow with a velocity shear of 0.01 s-1 is 1 m s-1. This is comparable, but not greater than, background speeds suggesting that it might influence the degree of isotropy by straining eddy structure in the horizontal direction.*"

- P8, l2 But how is $KZ$ computed here? Large values of $KZ = 10-1$ $m2s-1$ have been reported by Bluteau et al (2017), but they argue these high values are much more reliably estimated from the temperature spectra than the velocity spectra. As Bluteau et al (2016, JTECH, 33:713-722) argue integration methods are only robust if $\varepsilon \leq 10-6$ $m2s-1$. Author should consider this point carefully. I assume in all the processing that the author has used $\Gamma = 0.2$ ? While on average this may be globally true, the flow in Cook Strait seems very unusual with very high mean velocities and very high values of $Reb$ in Figure 11, for example. The point being that consistently here possibly $\Gamma \neq 0.2$ and it may be very misleading to assume that in the present observations – see Bluteau et al Fig 4.? So in Figures 7,8 and 9 is $KZ$ to be believed? There seems only one way to check this: independently compute $KZ$ from the temperature field, without any a priori assumption on the value of $\Gamma$.
*The Reviewer rightly picks up on one of the major themes in ocean turbulence – the efficiency of mixing – this is too big for this manuscript and dataset which focuses on the LOz/LT question. The Reviewer also picks up on the unusual nature of the flow with its high mean velocities. The changes I have made in response are to remove panel (b) of previous Fig. 10. (the Kz distribution) and replace the axes in Fig 7, 8 & 9 with 0.2eps/N2 and then expanded the Discussion. Given the bounds suggested by Bluteau et al 2017 there should still be meaning given the dynamic range observed. This enables the later discussion to be augmented as well as emphasised what future work is required. The revised text now all considers the related point made by Smyth et al. 2001, based on DNS of patches, which demonstrates the order of magnitude variability in LO/LT over the lifetime of the turbulent event.*

- P9, l7 The range of $Reb$ estimates is 2 orders of magnitude? Figure 11 suggests more than 4 orders of magnitude?

*Agreed, the original text was misleading and has now been clarified. It now states… "In the present Cook Strait data, the majority of Reb estimates exceed 100, with the peak of the distribution being around 5x1^04 two orders of magnitude with the peak of the distribution around 5x10^4. (Figure 11). However, maximal values exceed 10^7, which is primarily due to the small N."*

- Fig 12 suggests a very poor correlation between Lo and Lt – its log-log after all! *I accept the Reviewer's point that the best-fit distribution is centred on some widely spread data points. As noted by the Reviewer these variables are "independent" in the sense that they are derived from different components of the profile data. However, this level of variability is consistent with the spread of results of Wesson and Gregg (1994) all the more so because we calculate LT using the microstructure sensors allowing for a much smaller minimum lengthscale. Given that these data are at one limit of ocean energetics I believe we have to be careful about rejecting data because they don't conform to expectations. I have added material to the first subsection of the discussion on this point.*

- P11, l13 Maybe it simply means that the gamma is not 0.2, irrespective of the Re? *Is the Reviewer suggesting that the Kz is different enough to make the scaling argument not useful? Given that the scaling is linear in Gamma and the present homogenization time is 25 hours, this suggests that the Kz might be out by a factor of 5 say - i.e. homogenization takes 5 hours. The text is now amended to connect this point with that raised above around Gamma=0.2. It says "The Γ=0.2 "constant" is a clear point of contention in the literature (e.g. Bluteau et al. 2013; Mashayek et al. 2013). Bluteau et al. (2017) develops an approach that takes microstructure profiles and resolves the diffusivity "directly" fitting a model for dissipation of thermal variance to the convective-inertial subrange (i.e. lower wavenumbers than the dissipation scale). The Bluteau et al. (2017) analysis suggests that improved estimation of the thermal diffusivity indicates that the fixed mixing coefficient might underestimate mixing by a factor of 5 in the mean especially for the more turbulent events. Extending this by applying the Osborne diffusivity method sees an average diffusivity is around 0.04 m2 s-1 and exceeding 1 m2 s-1 (Figure 10b). One might expect a 300 m water column to then be homogenised in a time (L2/Kz=) 3002/1=25 hours, but this might be as little as 5 hours if the Bluteau et al. (2017) increased estimate of Kz were to hold. "*

- P14 line 10 where is the Hogg reference cited.? *Thanks for spotting this missing reference. This has now been included (Table 1 caption and Discussion).*

- The measurements themselves are presented with (too) many figures, but basic information is missing. Little is said, for instance, on the timing of the collected 34 profiles covering a very large period of time of 5 years. Processing of the microstructure data must be described or documented in a much more precise way than with sentences like "in the usual way" or "An approach".

  *As noted in Response to Reviewer #1, additional details on sampling and microstructure analysis are now included. The profiles come from only a short window of 12 days within this longer period. Also, the driver is strongly tidal so periods between measurements should be irrelevant especially as the separation is between scene-setting broad scale information in the earlier data collection (where previous work demonstrated consistency from year to year) and the later microstructure work.*

  *Details on the profiling timing is contained in Figure 6 . This was erroneously not specifically referenced but was talked about in the previous version. I thank the Reviewer for spotting this and have now augmented the Methods section. The figure has been moved forward to Fig 2 and the associated text now says "The timing of the profiles during the 2012 sampling is shown in Figure 2. Long periods of contiguous sampling is difficult because a vessel suitably manoeuvrable to conduct the experiments is prone to weather limitations. Sampling over three days in 2012 centered on periods spanning northward, turning and southward tidal flows (Figure 2)."*

*In addition, The number of figures has been reduced by one – removing the Kz distribution and combining the eps and Re_b figures.*

- Considering the fast flows in this region and the irregular topography, three-dimensional effects (e.g. horizontal advection) are anticipated and should be discussed.
*The Reviewer makes a good point. This partly overlaps with a comment by Reviewer 1 about the veracity of the LT in such fast flowing waters. The text has now been amended to include discussion on this at the end of the first subsection in the Discussion. I argue, however, that the topography is not irregular at least in the region where data were collected. It is actually, over the distance travelled in any one tidal cycle, reasonably "regular" in the sense that there are no major changes in channel orientation and no submarine ridges running transverse to the flow (Fig. 1c). Evidence of this (i.e. lack of cross-strait eddies) is contained in the Strait being considered something of a bioregional barrier limiting across-strait connectivity (Forrest et al 2009). This is now included in the Discussion which says "the Strait has been identified as a dividing line in terms of ecological structure (e.g. Forrest et al 2009). The implication is that there is not a great deal of across strait transport. This supports the focus of the present work on the vertical structure. Furthermore, over the time it takes to drift through the strait all vessel tracks tended to be on an axis aligned with the strait. Over these scales of time and space the strait itself is bathymetrically reasonable consistent. It remains to conduct a study that will adequately quantify across-strait mixing and the associated drivers."*

- Some of the (many) typos and formal problems to be fixed.
  - Page 3, line 4 : "velocity Sh" ! velocity shear ?  *fixed*
  - Page 3, line 8 : "Do we actually observe high dissipation rates?" *fixed*
  - Bottom of page 5 and first paragraph of page 6 : please fill the gaps "xxx" and "X" "Y". *fixed*
  - Please carefully check references : some of them are missing or unused (Gregg and Oszoy, 2002; Matter et al, 2003 or 2005;. . . ) *both now included.*
  - Figure 6 seems not to be cited / discussed in the text. *Thanks for spotting this – this has now been moved to Figure 2 and discussed at some length.*
  - Many figures are provided but with very little discussion. The ratio number of figures to Length of discussion seems to be rather low. *The revised Discussion is expanded to provide a more lengthy treatment of the data and the issues. In addition, the manuscript has been reduced by one figure.*

- In conclusion, the data are interesting but the manuscript should be better focused to avoid wild discussions of many details (e.g. on individual profiles taken at different (unknown) location and times) and come to solid conclusions.

*I thank the Reviewer for their suggestions and critique. The modified Discussion now focuses more specifically on the questions posed in the introduction and provides additional insight into the mechanical context and how this relates to the observed data shown in the figures. I do defend though the descriptions of the individual profiles which are there to serve as a context to talk about some of the actualities that can get lost when considering a scatter diagram covering many orders of magnitude and many hundreds of realisations. This helps the reader keep in mind that oceanic turbulence structure cannot be considered to be a series of disconnected experiments (as implied in Mater et al. 2013) but connected parts of a continuum. As noted above, to be fair these authors, in the later 2015 paper, do move their perspective to ocean scales and again this is where the present manuscript picks up the comparison. In regard*

*to the examination of selected individual profiles I took inspiration from the canonical Wesson & Gregg paper that included a small number of individual profiles to ground the later synthesis within the context of the source data. I note also that Mater et al 2015 look at selected patches. I certainly wouldn't argue that the entire manuscript be filled with such examination, but I believe it helps put the synthesis into better context. In addition, thanks in part to the points raised by the Reviewer, I believe the manuscript now comes closer to contributing to the "solid conclusions" they seek for questions at the forefront of ocean turbulence for some time now.*

**References**

[revised manuscript text omitted]

**Figures**

[Figure]

Figure 1 Location showing (a) New Zealand, and within this, (b) Cook Strait Narrows is bounded by Cape Terawhiti (CT) to the east and the headlands of the (shaded) Marlborough Sounds to the west and with the Cook Strait and Nicholson Canyons to the south (CSC and NC).

5    The 200 m (solid) and 400 m (dashed) depth contours are marked, as well as the shoal at Fishermans Rock (FR). ADCP moorings are marked with blue circles. The microstructure data come from profile regions V1 and V2.

[Figure]

Figure 2 Sampling conditions showing average N-S water column velocity between 60 and 100 m depth and wind speed - both filtered with an hourly low-pass filter. The bars show microstructure sample periods. These bars are expanded in the daily sampling relative to tidal conditions is shown in (b), (c) and (d).

5

[Figure]

Figure 3 profiler drop speed from a number of example profiles (time and profile number on top).

Figure 2 profiler drop speed from a number of example profiles.

[Figure]

Figure 4 velocity data from eastern side of strait showing (a) east-west, (b) north -south, (c) vertical velocities and (d) backscatter amplitude.

[Figure]

Figure 5 Temperatures from near-bed (Ea, Figure 1) and satellite-derived sea surface temperature (SST). The arbitrarily-scaled spring-neap envelope is along the base of the panel.

5  Figure 4 Temperatures from near-bed (Ea, Figure 1) and satellite-derived sea surface temperature (SST). The arbitrarily-scaled spring-neap envelope is along the base of the panel.

[Figure]

Figure 65 Temperature and salinity ($S_p$) from the seabed sensors on the moorings with profile data overlain (see text for details) and $\sigma_T$

contours.  The arbitrary-scaled  histogram on the right-hand margin shows the distribution of satellite-derived sea surface

temperature (SST).

[Figure]

Figure 7̶7̶ Profile A from day 132, 2012 showing (a) temperature, salinity and buoyancy frequency squared ($N^2$) squared, (b) $L_T$ and $\varepsilon$ and (c) proxy for vertical diffusivity $0.2\varepsilon/N^2$Kz. Note the profile extends over the full depth of the water column.

[Figure]

Figure 88 Profile B from day 137, 2012 showing (a) temperature, salinity and buoyancy frequency squared (N²), (b) $L_T$ and ε and (c) proxy for vertical diffusivity 0.2ε/N². Note the profile extends over the full depth of the water column. showing (a) temperature, salinity and buoyancy frequency (N²) squared, (b) $L_T$ and ε and (c) vertical diffusivity $K_z$. Note the profile extends over the full depth of the water column.

[Figure]

5    Figure 99 Profile C from day 143, 2012 showing (a) temperature, salinity and buoyancy frequency squared ($N^2$), (b) $L_T$ and $\varepsilon$ and (c) proxy for vertical diffusivity $0.2\varepsilon/N^2$. Note the profile extends over the full depth of the water column. showing (a) temperature, salinity and buoyancy frequency ($N^2$) squared, (b) $L_T$ and $\varepsilon$ and (c) vertical diffusivity $K_z$. Note the profile extends over the full depth of the water column.

[Figure]

[Figure]

Figure 10̶1̶0̶ Distributions of (a) dissipation rate ε and (b) buoyancy Reynolds number, Re_b.inferred vertical diffusivity K_z.

[Figure]

[Figure]

5  **Figure 11 Distribution of buoyancy Reynolds number, Re_b.**

[Figure]

Figure 11 12 Scatter plot of $L_T$ vs $L_{Oz}$ colour-coded by depth. Lines for unity and for $L_{Oz}=0.8L_T$ are shown and associated histograms of length-scales are shown also. Averages were calculated in log10 space and 0.5 m was considered a lower-bound for $L_{Oz}$.

[Figure]

Figure 1213 The comparison of the $L_T/L_{Oz}$ ratio as a function on non-dimensionlised $L_T$. The present data set (circles with +/- 1x standard deviation as error bars) is superposed on top of synthesized results following Mater et al (2015).

[Figure]

Figure 1314 Comparison of dissipation resolved from the L$_{Oz}$ with the direct measure of ε.  Averages were calculated in log10 space for ε and all length-scale-based estimates in that bin were collated.  Here the average and average+ 1 standard deviation are shown with a circle-line pair and the average in log10 space is shown as a square.  The averages excluded outliers in the surface water as described in the text.

---

## Referee Report (RR1)

**Comments on Revised version "Turbulent length scales in a fast-flowing, weakly-stratified Strait: Cook Strait, New Zealand.**

I find the revised manuscript much improved over the first draft, and the author has responded carefully and fully to the various questions I initially raised. I would recommend publication and have only minor final suggestions listed below.

On the general question of the use of the Thorpe scale  $L_T$ , it is obtained from a one-dimensional re-sort of the vertical density profile. As the author is aware, this has its limitations because the scale can exceed the depth. However, there us a deeper fundamental problem. One interpretation of the important ideas proposed by Peltier and Caulfield (Annual Review Fluid Mechanics, 35, 2003), is that one must do a three-dimensional resort to get a true measure of the energy and turbulence locked in a stratified fluid. That can't be done from a single microstructure profile, and hence its hard to interpret what is the meaning of a quantity like the Thorpe scale.

The Bluteau et al 2017 paper also stresses that the mixing efficiency can also be considerably less than 0.2, depending on the local flow field. Note also the paper by Salehipour and Peltier (JFM, 775, 2015) demonstrates the limitations of the usual 0.2 efficiency assumption in the Osborn model - see their Fig 2.